# Self-Regulatory Neuronal Mechanisms and Long-Term Challenges in Schizophrenia Treatment

**DOI:** 10.3390/brainsci13040651

**Published:** 2023-04-12

**Authors:** Agnieszka Markiewicz-Gospodarek, Renata Markiewicz, Bartosz Borowski, Beata Dobrowolska, Bartosz Łoza

**Affiliations:** 1Department of Human Anatomy, Medical University of Lublin, 20-090 Lublin, Poland; 2Department of Neurology, Neurological and Psychiatric Nursing, Medical University of Lublin, 20-093 Lublin, Poland; renatamarkiewicz@umlub.pl; 3Students Scientific Association at the Department of Human Anatomy, Medical University of Lublin, 20-090 Lublin, Poland; bartosz.borowski@interia.pl; 4Department of Holistic Care and Management in Nursing, Medical University of Lublin, 20-081 Lublin, Poland; beatadobrowolska@umlub.pl; 5Department of Psychiatry, Medical University of Warsaw, 02-091 Warsaw, Poland; bartosz.loza.med@gmail.com

**Keywords:** schizophrenia, neural plasticity, synaptic plasticity, brain stimulation, therapeutic methods

## Abstract

Schizophrenia is a chronic and relapsing disorder that is characterized not only by delusions and hallucinations but also mainly by the progressive development of cognitive and social deficits. These deficits are related to impaired synaptic plasticity and impaired neurotransmission in the nervous system. Currently, technological innovations and medical advances make it possible to use various self-regulatory methods to improve impaired synaptic plasticity. To evaluate the therapeutic effect of various rehabilitation methods, we reviewed methods that modify synaptic plasticity and improve the cognitive and executive processes of patients with a diagnosis of schizophrenia. PubMed, Scopus, and Google Scholar bibliographic databases were searched with the keywords mentioned below. A total of 555 records were identified. Modern methods of schizophrenia therapy with neuroplastic potential, including neurofeedback, transcranial magnetic stimulation, transcranial direct current stimulation, vagus nerve stimulation, virtual reality therapy, and cognitive remediation therapy, were reviewed and analyzed. Since randomized controlled studies of long-term schizophrenia treatment do not exceed 2–3 years, and the pharmacological treatment itself has an incompletely estimated benefit-risk ratio, treatment methods based on other paradigms, including neuronal self-regulatory and neural plasticity mechanisms, should be considered. Methods available for monitoring neuroplastic effects in vivo (e.g., fMRI, neuropeptides in serum), as well as unfavorable parameters (e.g., features of the metabolic syndrome), enable individualized monitoring of the effectiveness of long-term treatment of schizophrenia.

## 1. Introduction

Neuroplasticity is a phenomenon studied mainly through synaptic plasticity, in which—under the influence of external and internal stimuli—functional and morphotic changes occur during the transmission of signals in the nervous system [1,2,3,4,5]. Because of this mechanism, more attention is being paid to studies that analyze the neuroplastic influence of various environmental interactions, including therapeutic activities. While studies of the plasticity of excitatory (e.g., glutamatergic) synapses are well documented, studies of inhibitory (e.g., GABAergic) synapses are still inconclusive [6,7]. Undoubtedly, both excitatory (postsynaptic membrane becomes depolarized) and inhibitory (postsynaptic membrane becomes hyperpolarization) synapses exhibit their neuroplastic potentials, maintaining homeostatic balance in neuronal circuits [6,7]. 

Synaptic plasticity is a fundamental feature of the brain and is defined as a permanent change in the response of neurons induced by stimuli from the external environment. At the cellular level, the main mechanism for changing synapse strength is synaptic plasticity, which operates according to Hebb’s rule of simultaneous activation of postsynaptic and presynaptic neurons [3]. Synaptic plasticity is crucial for cognitive processes, i.e., learning, memory retention, and recall, occurring at various stages of life in response to environmental or internal stimuli [1,8,9,10,11,12,13]. Specific forms of learning are thought to correspond to specific changes in neuroplasticity in different areas of the brain. For instance, hippocampal and medial temporal lobe synapses encode spatial learning [1,14], striatal synapses encode habit formation [10,11], motor cortex synapses encode motor sequence learning [12,15], and sensory cortex synapses encode perceptual learning [13]. 

The integrity of different brain areas, which is supported by synaptic plasticity, is a prerequisite for proper brain maturation and proper formation of neuronal circuits. Two major opposite forms of synaptic plasticity, long-term potentiation (LTP) and long-term depression (LTD), associated with changes in AMPA (α-amino-3-hydroxy-5-methyl-4-isoxazolepropionic acid) and NMDA (N-methyl-d-aspartate) receptors, are usually present in parallel in synapses in response to different patterns of activation [1,16,17,18]. Consequently, the result of properly functioning neuronal circuits is the proper reception and adequate interpretation of external and internal stimuli (cognitive, motor, sensory) [19,20]. The neuroplasticity network (Figure 1) includes various mechanisms that allow the brain to change through growth and reorganization [21]. These changes range from individual neurons making new connections and learning new abilities to systematic, complex adjustments like cortical remapping.

Since schizophrenia is a chronic disease with an unclear etiology (e.g., neurodevelopmental, genetical, immunological, psychological, biochemical), many hypotheses of impaired synaptic plasticity are formulated, such as excessive synaptic pruning, [22], impaired functioning of neuronal circuits [23,24], abnormalities in the postsynaptic protein signaling complex [25,26,27,28], deficits in function-specific gray matter areas (especially prefrontal, frontal, and temporal) [29,30,31,32], disturbed excitation, and GABAergic inhibition [33].

Despite the fact that the concepts of impaired synaptic plasticity in schizophrenia are only preliminary, exploratory studies are being continued to look for methods that could regulate adequately neuronal circuits. Since neither the available treatments for schizophrenia nor the understanding of their basic mechanisms are sufficient for their practical generalization, the concept of modifying neural circuits is attractive as a heuristic, biological, and clinical target. While in the treatment of exacerbations of schizophrenia (psychotic symptoms) antipsychotics are usually administered and then must be administered for years to prevent relapses, the improvement in negative symptoms, patients’ quality of life, and the achievement of their strategic social goals do not find consensual paths in therapeutic algorithms [34,35]. Moreover, it has been postulated that antipsychotic medication can impair long-term potentiation and cognition by altering dopaminergic transmission [36].

Since randomized controlled studies of schizophrenia treatment do not exceed 2–3 years, and the pharmacological treatment itself has an incompletely defined benefit-to-risk ratio, we reviewed treatment methods based on the paradigm of long-term neural self-regulatory and neural plasticity mechanisms.

The abnormal activity of the neural circuits is due to the clinical diagnoses of the subjects, which cause limitations in their social functioning to varying degrees. The main goal of comprehensive treatment is to restore these pathological activities and normalize their function [37,38,39]. Based on various forms of therapy, rehabilitation standards, and procedures, improving treatment outcomes will be a new challenge for medical and therapeutic stuff.

## 2. Neurofeedback

Among the various forms of neurotherapy used for patients with mental disorders, biofeedback, a non-invasive method that allows patients to voluntarily control their mental and physiological functions based on feedback from on neurophysiological activity, is gaining popularity [40]. Depending on the type of brain-computer interface (BCI) used, we distinguish different forms of biofeedback: GSR-BF (biofeedback based on analysis of skin-galvanic response), EEG-BF (biofeedback based on analysis of brain activity), EMG-BF (biofeedback based on analysis of muscle response), and HRV-BF (biofeedback based on analysis of heart rate profile) [41]. A growing number of reports indicate that biofeedback therapy has positive results in patients with anxiety disorders, depressive disorders, suicidal tendencies, bipolar disorder, and schizophrenia [41,42,43,44]. There are various physiological concepts of how biofeedback therapy works [45,46,47]. They refer to neuroplastic and behavioral theories, cognitive training, conditioning, or modulations of one’s own neural activity. Biofeedback training can be viewed as a form of external influence that uses specific exercises to modify the structure and function of neuronal networks through learning and memory (internal influence) [45].

In 2013, combining functional MRI (fMRI) with biofeedback training confirmed the possibility of conscious control of brain activity in patients with a diagnosis of schizophrenia [48,49]. Repeated stimulation (repetitive training) has been proven to have a positive effect not only on stimulus-response components but also on changing the intensity of interneuronal connections and increasing the number of synaptic connections [50]. As shown, regular training contributes to many physiological changes occurring in the body, including changes in the activity of brain-derived neurotrophic factor (BDNF), which is associated with an increase in neuronal gene expression and reorganization of synapses [51]. The relationship between physical and cognitive activity and BDNF levels appears to be bidirectional [52,53]. This is supported by studies that show a correlation between BDNF levels and its signal in many neuropsychiatric diseases [47], a correlation between reduced BDNF levels and the development of schizophrenia [53,54], and a negative correlation between negative and total PANSS scores and serum BDNF levels [52]. Since the neuropsychological effects of BDNF can be monitored based on peripheral serum levels [55], as can nerve growth factor (NGF) and the neurotrophins NT-3 and NT-4/5 (proteins that promote the formation of synapses), it seems reasonable to attribute positive significance in rehabilitation to the biofeedback method. A series of systematically implemented biofeedback training sessions results in BDNF being involved in neuronal function [56], growth and differentiation of synapses, regulation of neuronal circuits [57], and formation of memory pathways [53,58].

## 3. Vagus Nerve Stimulation (VNS)

The vagus nerve is an X cranial nerve originating developmentally from the fourth and fifth pharyngeal arches. It is mixed nerve composed of approximately 100,000 axons, 80% of which are bare nonspecific and specific centrifugal visceral (motor and secretory) and centripetal (visceral and somatic) fibers. The vagus nerve plays an important role in regulating metabolic homeostasis and plays a key role in the neuroendocrine-immune axis [59].

VNS is one of the best studied physical neuromodulatory methods. Its effectiveness has been proven in the treatment of drug-resistant chronic and/or recurrent depression [60] and medically refractory epilepsy [61]. Based on available reports related to the efficacy of VNS in treating depressive and biological modulations in cortical-subcortical networks, which are also involved in the pathogenesis of schizophrenia, it can be concluded that VNS may be an additional adjunctive treatment option for patients with a diagnosis of schizophrenia [62]. VNS can most often be performed surgically. It is best achieved by surgically implanting a stimulating, conductive wire around a nerve in the neck [63]. The wire is periodically stimulated by a generator implanted in the left chest wall. Stimulation is directional, minimizing potential side effects, and stimulus parameters (e.g., intensity) can be programmed by the doctor. However, a new non-surgical method of VNS known as tVNS has recently been developed. The tVNS device will be used to stimulate the auricular branch of the vagus nerve using a bipolar electrode attached to the skin of the left ear shell [64].

The first pilot study conducted to investigate the feasibility, safety, and efficacy of non-invasive transcutaneous vagus nerve stimulation (tVNS) demonstrated that this procedure is well tolerated by patients [62]. In an experimental study conducted on a group of 20 patients with schizophrenia, VNS was shown to significantly reverse hippocampal hyperactivity, mesolimbic dopaminergic dysfunction, and schizophrenia-like symptoms, including cognitive deficits [62,65].

Interest in the vagus nerve as a facilitator of plasticity stems from early studies that pointed to the vagus nerve in enhanced memory consolidation. In a landmark study, Clark et al. (1994) showed that electrical stimulation of the vagus nerve immediately after training improved memory, providing a direct link between vagus nerve activity and modulation of CNS function [66]. The anatomical and functional connectivity of the vagus nerve provides a clear basis for its effects on the CNS. Although it is most often recognized for its descending spicules to the viscera, >80% of the vagus nerve consists of afferent connections that terminate in the nucleus tractus solitarius in the brainstem [67,68]. Electrical stimulation of the vagus nerve stimulates cholinergic forebrain and sinusoidal noradrenergic site activity and subsequently causes the release of neuromodulators in the cortex [69,70]. Decreased noradrenergic or cholinergic transmission blocks the effects of VNS in the CNS [71]. Both neuromodulatory systems are key substrates in the expression of cortical plasticity [71,72], providing the rationale by which VNS combined with rehabilitation can improve recovery.

## 4. Repetitive Transcranial Magnetic Stimulation (rTMS)

Transcranial magnetic stimulation is a sequence of repeated, short, and highly focused magnetic pulses used to stimulate brain cells. The above method has been known and used for many years. It is the youngest electrophysiological method allowing non-invasive and non-painful stimulation of the central and peripheral nervous systems [73]. Considering the central nervous system (CNS), superficial structures, i.e., the cerebral cortex and cerebellum, as well as the medulla oblongata at the level of the great aperture, are effectively stimulated [74]. Moreover, stimulation with rTMS can alter brain metabolic activity, neuronal plasticity, local brain function, and functional connectivity between different brain regions [75]. It is known to have therapeutic effects in several neuropsychiatric disorders, including major depression, conversion disorder, schizophrenia, and obsessive-compulsive disorder [76,77]. Available studies suggest that in patients with a diagnosis of schizophrenia, rTMS at low frequency, i.e., about 1 Hz, is used to suppress auditory hallucinations, and rTMS at high frequency, i.e., 10 Hz, is applicable for negative symptoms and may have a satisfactory effect [78,79]. In addition, recent studies have shown that low-frequency rTMS over the left temporo-parietal ventricle reduces the occurrence of auditory verbal hallucinations in schizophrenia and that it can be an effective treatment for patients through selective neural modulation underlying psychiatric symptoms [80]. Another study performed by Guan et al. (2020) notes that the use of higher rTMS frequencies, i.e., 20 Hz, for a period of 8 weeks can significantly improve the direct memory score, which was correlated with a decrease in the arousal factor in patients with schizophrenia [81]. The differing reports on the rTMS frequencies used are due to the varying triggering and underlying mechanisms of schizophrenia. Moreover, the intensity of stimulation itself is enough to affect the rTMS-induced changes in cortical excitability. Intensity is defined as the percentage of an individual’s resting motor threshold (MT) that is the starting point for the stimulation required to produce motor evoked potential [82]. Stimulation at a higher intensity than a patient’s resting MT is results in more long-term effects, whereas stimulation below the MT threshold may result in less spectacular effects [83]. 

TMS is performed by a technician or physician. It is an outpatient procedure, so it can be performed in an outpatient clinic. A standard TMS examination involves qualifying the patient and explaining the test. Then, the motor threshold (MT) is determined separately for each cerebral hemisphere, and the motor evoked potential (MEP) and the central silent period (CSP) are recorded. Root stimulation and neurographic examination of relevant peripheral nerves, including F-wave recording, are also performed [73]. 

Currently, problems with the treatment of schizophrenia are a significant concern. The use of pharmacological agents causes many side effects and is not always applicable to all symptoms of the disease, so alternative forms of treatment are being sought [31]. Increasing evidence suggests that rTMS may contribute to the relief of positive and negative symptoms of schizophrenia [84]. The negative symptoms of schizophrenia, including alogia, avolition, anhedonia, and affective flattening, are associated with attention dysfunction [85]. These are most likely associated with decreased activation, primarily within the left prefrontal cortex [86]. Additionally, they often remain refractory to neuroleptic drugs and are thus more difficult to treat, so innovative therapeutic approaches are being sought to increase activation in this brain region [87]. 

A study conducted by Cohen et al. (1999) included a relatively small group of patients with chronic schizophrenia (*n* = 9). The above pilot study was designed to evaluate whether direct application of prefrontal cortex rTMS could improve negative symptoms or cognitive impairment and whether rTMS could modify deficits in prefrontal cortical activity in chronic schizophrenia [78]. A decrease in negative symptoms was observed, and significance was noted on the PANSS negative symptoms subscale. Another study by Sachdev et al. (2005) suggests a potential role in treating the negative syndrome of schizophrenia [79]. However, the authors of the above paper note that this is a relatively small study and warn against drawing hasty conclusions. Additionally, the rTMS therapy they used (4 weeks) confirms that it is a feasible and safe method that does not contribute to the worsening of positive symptoms in this group of patients. Hajak et al. (2004) also confirm a statistically significant improvement in negative symptoms in patients with schizophrenia (using high frequency rTMS) [88]. However, interestingly, in the above study, they also find a worsening of positive symptoms compared to baseline results. High frequencies of rTMS can increase cortical excitability and modulate the release of dopamine, the lack of which in the prefrontal cortex can cause negative symptoms of schizophrenia [85]. In addition, rTMS may modify the expression of glutamic acid decarboxylase, which is a synthetic γ-aminobutyric acid (GABA) precursor enzyme, which may be important because negative symptoms scores are inversely related to benzodiazepine binding in the medial frontal region [89].

## 5. Transcranial Direct Current Stimulation (tDCS)

Transcranial direct current stimulation is a non-invasive neuromodulatory technique [90] that is increasingly used to treat psychiatric disorders. Its main mechanism of action involves the use of subthreshold modulation of neuronal membrane potentials, which alters the activity of the cerebral cortex depending on the direction of current flow through target neurons [91]. The flow of current is accompanied by biological effects of the electric field, i.e., changes in the level of neurotransmitters, changes in glial cells, small vessels, and/or modulation of inflammatory processes. The basis for these changes is the Hebbian theory, which states, “When the axon of cell A is close enough to excite cell B and is repeatedly involved in activation, growth processes or metabolic changes take place in one or both cells, so that both the output of cell A, and the excitatory capacity of cell B are increased.” Thus, if presynaptic and postsynaptic neurons are synchronized and active, then synaptic potentiation occurs, while if there is no mutual synchronization, then “the neurons disconnect” [92]. It is assumed that neuroplasticity is mediated by long-term potentiation, which depends on postsynaptic calcium levels, involving N-methyl-D-aspartate (NMDA) [93] and α-amino-3-hydroxy-5-methyl-4-isoxazolopropionic acid (AMPA) receptors [94]. 

Moreover, to be able to achieve the synchronization effect, it is necessary to use a constant electric current generated by a generator connected to two electrodes (anode and cathode), which should be placed at specific locations on the head. Part of the current with fixed parameters (standard: 1–2 mA) flows through the scalp, skull, and cerebrospinal fluid, while part changes the resting membrane potentials of neurons and increases the probability of depolarization or hyperpolarization (without inducing action potentials) [95]. Although the effects of tDCS occur mainly under the electrodes, its direct or indirect effects have a significant impact on distant neuronal networks [96,97].

Schizophrenia as a chronic neuropsychiatric disorder is characterized by delusions, hallucinations, disorganization of speech and behavior, and reduced emotional expression [98]. The deficits that result from the chronicity of the disease are indicative of its severity [99]. Since about 20% of patients are refractory to treatment [100], new NIBS (noninvasive brain stimulation interventions) techniques including rTMS and tDCS have been introduced into rehabilitation interventions [101,102]. The starting point for tDCS therapy was the neuroimaging results: cathodal stimulation of the hyperactive left temporoparietal area (reducing auditory hallucinations) [103] and anodal stimulation of hypoactive frontal areas (targeting mainly negative symptoms) [104]. The effect of tDCS in RCTs (randomized clinical trials) on auditory hallucinations and negative and positive symptoms in schizophrenia showed that the use of tDCS twice a day (therapy with similar parameters) has a positive effect on reducing auditory hallucinations using the Auditory Hallucination Rating Scale (AHRS). After just 1 week of twice daily tDCS use, patients in the study group showed a significant 31% decrease in AHRS scores and a 38% improvement 3 months later [105]. These results were also confirmed by Mondino et al., (2015) [106]. In another study conducted in hospitalized patients with refractory schizophrenia, the total AHRS score in RCTs decreased by 21.9% after 4 weeks of tDCS therapy [107]. Both the frequency of auditory hallucinations and their duration were reduced. Although the clinical significance of the study is pilot, the results obtained seem encouraging. The use of this form of therapy is further supported by the fact that antipsychotic drugs are mainly effective in reducing the positive symptoms of schizophrenia, while their effectiveness in treating negative symptoms is limited [108].

## 6. Virtual Reality (VR) Therapy for Patients with Schizophrenia

Virtual reality is a three-dimensional image of (parallel) reality created by computer. VR can represent various objects, subjects, or events. The use of VR applications has received a lot of attention in recent years, including in the clinical treatment of people with various mental health disorders. Although VR technology shows great promise, the required hardware is relatively expensive, and the software development process is still in the testing stages. Nevertheless, the disadvantages are not deterring laboratories from researching the potential of VR in cognitive rehabilitation [109]. The therapeutic interventions used can work by helping patients reacquire cognitive abilities through repetitive, systematic, hierarchical, restorative cognitive stimulation or by teaching alternative compensatory strategies that focus on actual task performance [110]. 

Schizophrenia is characterized by disturbances in cognitive aspects, and these problems specifically involve attention, memory, and executive functions [111]. According to the WHO, schizophrenia is a common mental illness, affecting some 21 million people [112]. People diagnosed with schizophrenia may experience significant difficulties in social cognitive functioning, including problems with understanding actions, emotions, social perception, and/or empathy [113]. Some of the psychotic symptoms, i.e., hallucinations and delusions, are partially alleviated by antipsychotic drugs, but social impairments are a significant problem, complicating the path to health stability/recovery [114]. In general, the use of antipsychotic drugs has reduced effectiveness on the negative symptoms of schizophrenia. These symptoms are associated with negative effects on the patient’s functional status and quality of life. The clinical expression of negative symptoms is less pronounced compared to positive symptoms, as they may be masked by positive symptoms and may coexist with or be confused with affective symptoms or cognitive disorders [114,115]. The above dysfunctions affect patients’ daily functioning. 

Freedman et al. (2016) proposed applying the VR CBT model to persecutory delusions derived from inconsistent threat beliefs [116]. In their study, they used exposure therapy for the first time in VR and CBT sessions to treat delusions. Comparing VR exposure therapy with VR exposure modified with cognitive therapy elements in six VR scenarios, they proved that VR CBT was more effective than VR exposure (by 22%). The above results were also confirmed by the team of Pot-Kolder et al. (2018), who further demonstrated that VR CBT can significantly reduce paranoid thoughts in patients with a diagnosis of schizophrenia [117].

Serious games and the use of virtual reality allow for the creation of immersive environments that allow for more effective and long-lasting recovery while improving quality of life [118]. The use of virtual reality can enhance traditional evaluation/rehabilitation programs, strengthening the results achieved, both cognitively and socially, contributing to the legitimacy of the psychosocial integration [119].

## 7. Cognitive Remediation Therapy

Although there is already encouraging experience for psychosocial interventions during the period of immediate exacerbation of schizophrenia symptoms—rather heterogeneous and sometimes contradictory—here we refer only to long-term cognitive remediation programs [120]. Simple social-cognitive trainings may have a positive effect on emotion recognition, mental state attribution, or social perception, but not generally on psychotic symptoms or overall level of social functioning [121]. In the case of more complex cognitive remediation therapies (CRTs) for schizophrenia, there is a noticeable improvement in the attention domain in proportion to the duration of treatment, especially if the technique of a “bridging” discussion group is used as an aid, having the potential to generalize and consolidate the results [122]. However, so far, longitudinal CRT studies have illustrated only a small-to-moderate association between cognition and community functioning (including successful social and job relationships) [122]. Overall, the results of CRTs are heterogeneous; however, it may be possible to prove basic relationships among cognitive trainings, rehabilitation, neuropeptide activity, and clinical improvement [123].

## 8. Pharmacotherapy and Neuroplasticity

Antipsychotic drugs cause numerous synaptic changes, and the assessment of their long-term neuroplastic effects is ambiguous. The neuroplastic effects would be what distinguishes first-generation antipsychotics from second-generation (atypical) drugs [124]. Atypical antipsychotics exert measurable neuroprotective effects (preventive, restorative), mediated via multiple molecular mechanisms and often in a dose-dependent manner, and may play a core role in ameliorating the neurodegenerative effects of psychosis [125].

No one is against the efficacy of antipsychotic medications for relapse prevention in schizophrenia in the short to medium term [34]. However, over 3 years of treatment follow-ups, there are only naturalistic cohorts with indirect data on the long-term neuroplastic consequences of antipsychotic use. The development of supersensitivity of the dopamine D2 receptors may lead to concern over the cumulative effects of antipsychotics on physical health and brain structure [125]. Effectiveness of low-dose or gradual discontinuation strategies warn against chronic antipsychotic use. Nowadays, the lack of randomized long-term studies makes it impossible to differentiate brain changes resulting from the schizophrenia from secondary effects, including the impact of antipsychotic drugs. The right question in this situation is what long-term treatments allow for the maintenance of neuroplasticity. We observed that combining stable pharmacotherapy for schizophrenia with intensive rehabilitation enables specific changes in the profile of neuropeptides and the achievement of long-term clinical goals of treatment [123].

A meta-analysis suggested that peripheral BDNF levels are positively correlated with some atypical antipsychotics (but not typical ones, like haloperidol), regardless of proper clinical response [126,127]. Antipsychotics may promote neurogenesis and affect synaptic architecture, the shape of dendritic spines, which mediates the clinical and cognitive effects of these agents [127]. 

## 9. Detection and Modulation of Neuroplasticity Effects 

Only those methods supporting modulation that are clinically applicable in the treatment of patients with schizophrenia are listed in Table 1.

## 10. Discussion

Long-term observations suggest that constant receptor interactions with antipsychotic drugs may help prevent further exacerbations of schizophrenia, while worsening the prognosis of achieving recovery, defined as meeting the criteria of symptomatic but also functional remission [155]. Seven years after the first psychotic episode, patients gradually reducing the dose of antipsychotics experienced twice the recovery rate compared to the maintenance group (40.4% vs. 17.6%, odds ratio of 3.49, *p* = 0.01) [155].

Most of the long-term cohort studies found a decrease in efficacy during chronic treatment with antipsychotics that cannot be explained by increasing non-adherence, because compliance is also an appropriate target for treating schizophrenia [156]. In neuroimaging and neurofunctional studies, clinical consequences of brain functioning as a potential result of chronic antipsychotic exposure, likely from dopaminergic hypersensitivity, is proven by irreversible neurobiological changes [156]. 

There is some evidence suggesting that atypical antipsychotics may partially improve cognitive functions; however, at the same time, they can induce serious, long-term metabolic adverse effects, such as obesity, dyslipidemia, and type 2 diabetes, which are linked to impairments in cognition [157]. Insulin resistance syndrome has been found to be directly related to cognitive dysfunctions in the context of obesity and type 2 diabetes [158]. Insulin resistance has been shown to be associated with impaired hippocampal synaptic plasticity and memory [159], as well as impaired neurogenesis [160].

However, even critics of the maintenance treatment do not interpret these results as the inevitable choice of a discontinuation strategy [34,156]. The proper question of this dilemma would be—if schizophrenia is characterized by impaired neuroplasticity and reduced expression of neurotrophic molecules promoting growth, survival, and differentiation of neurons—how these processes can be promoted without compromising treatment effectiveness, especially in the maintenance phase.

The problem is that if the current randomized trials of schizophrenia treatment cover 2- to 3-year periods only, the recommendations regarding the maintenance treatment phase are rather theoretical projections of the principles developed in the acute and subacute phases of treatment [34]. Moreover, the development of alternative treatment strategies for schizophrenia is constrained by dramatic prognosis of the mortality risks of patients on and off medication [161]. Difficulty in changing the approach may also result from the prejudice that non-pharmacological treatments are in fact limited to trivial “support,” with potential that is disproportionate to that of drugs. Such prejudices can only be changed by scientific facts [156].

In order to plan optimal therapy, it is necessary to characterize the neuroplastic properties of individual antipsychotic drugs and to determine their balanced, long-term neuroplastic potential. These types of findings are currently very preliminary and equivocal, although some results already differentiate main generations of antipsychotics (typical vs. atypical) [162]. Accordingly, all treatments should be redefined in terms of their neuroplastic potential and how to monitor these effects in vivo. This is consistent with most of the non-pharmacological techniques listed in this article, such as rehabilitation therapy or cognitive remediation therapy, whose effects are always spread out over time, gradually accumulating positive effects of neural plasticity.

For this reason, the search for therapeutic methods alternative to or compatible with pharmacotherapy and most likely combined with one another in terms of long-term stimulation of neuroplastic effects seems to be a promising direction for the development of schizophrenia treatment strategies.

Therefore, there is an urgent need for a specific type of long-term and multi-arm studies investigating the integration of pharmacological and non-pharmacological methods in pursuit of two main goals: relapse prevention and functional outcomes, with the identification of specific biomarkers of neuroplasticity, demonstrating proper internal and external validity, as well as acceptable levels of predictive accuracy (i.e., sensitivity and specificity) that justify their clinical usefulness.

## 11. Conclusions

Since randomized controlled studies of long-term schizophrenia treatment do not exceed 2–3 years, they may not fully capture the long-term effects of patient treatment, and the pharmacological treatment itself has an incompletely estimated long-term benefit-risk ratio. Thus, treatment methods based on other paradigms, including neuronal self-regulatory and neural plasticity mechanisms, should be considered.Neural plasticity can be a common platform for evaluating effective treatment of schizophrenia, defined as meeting the sustainable criteria of symptomatic and functional remission using both pharmacological and non-pharmacological treatment methods.There are increasingly more methods available for monitoring neuroplastic effects during schizophrenia therapy (e.g., fMRI, neuropeptides in serum), as well as unfavorable parameters (e.g., features of the metabolic syndrome). The use of these methods enables individualized monitoring of the effectiveness of long-term treatment of schizophrenia; however, the availability and cost of some alternative treatments (e.g., neurofeedback, rTMS, neuropeptides in serum, fMRI) may limit their use among patients and may not be available in all centers and treatment conditions.The effectiveness of alternative treatments may vary across individuals and may not be effective for everyone.The potential adverse effects of some of the alternative treatments, such as transcranial magnetic stimulation and vagus nerve stimulation, may limit their use in some patients.

## Figures and Tables

**Figure 1 brainsci-13-00651-f001:**
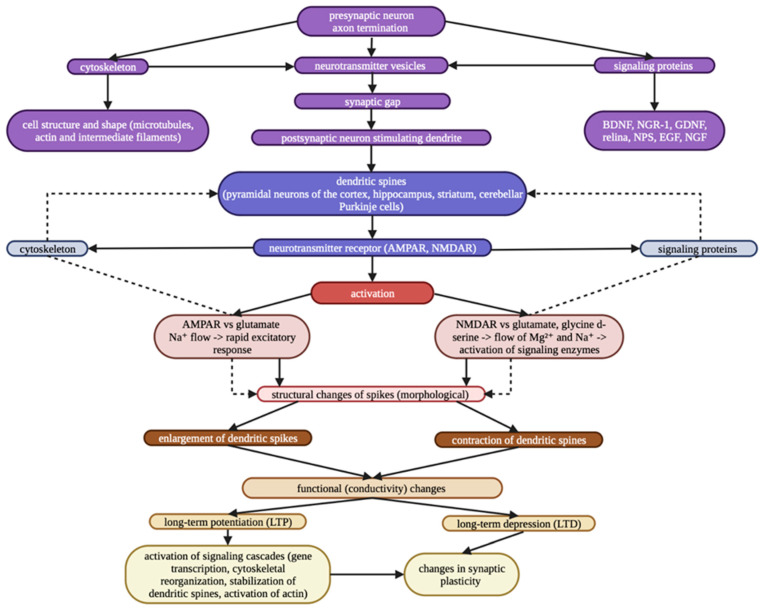
Neuroplasticity network.

**Table 1 brainsci-13-00651-t001:** Characteristics of methods supporting modulation in patients diagnosed with schizophrenia.

Method Type	Characteristic	Ref.
Functional magnetic resonance imaging (fMRI)	Unveils changes in the blood perfusion of those brain parts that are active during simple task management, as they require increased oxygen intake. Most fMRI studies involving patients with schizophrenia have directly examined working memory (BOLD effect), e.g., after cognitive remediation or after an intensive computerized training, exploring patterns of arousal in the measurements. Finc et al. (2017) describes patients dealing with experimental tasks that were leading to the positive reorganization (adaptation) of the brain process.	[128,129,130,131]
Transcranial magnetic stimulation (TMS)	Is a non-invasive method generating magnetic stimulation, which influences the cortex of the brain through the patient’s scalp. Among various cortical regions, TMS activates motoneurons and their tracts, which can be detected, making the model of CNS integrity. TMS can be used in the diagnosis and assessment of the severity of neurodegenerative diseases like schizophrenia. TMS is not only a diagnostic measure but can also be applied as a stimulation in drug-resistant neuropsychiatric disorders, including schizophrenia.	[132,133,134,135]
Magnetoencephalography (MEG)	Detects magnetic fields caused by the synchronous activity of neurons and provides high-quality spatiotemporal maps of electrophysiological activity. The new version of MED offers increasingly higher resolution of the map of neuronal activity and the ability to conduct research in motion (optically pumped magnetometers; OPM-MEG). MEG studies in schizophrenia are focused on the whole-brain resting state activity, auditory encoding processes, and functional connectivity. MEG in schizophrenia, as in diseases associated with brain damage, could monitor the neuronal healing process.	[136,137,138]
Pluripotential stem cells	By analogy with the experiment of Burbulla et al. (2021), in which the development of neurons with pluripotent stem cells was induced, which enabled the analysis of abnormalities in neuroplastic effects in patients struggling with Niemann-Pick type C syndrome, attempts are also made to use this method in patients with schizophrenia.	[139,140,141]
Positron emission tomography (PET)	Is a functional imaging technique that uses radiotracers to visualize and measure metabolic processes. PET imaging of synaptic protein 2A (SV2A) in schizophrenic patients presented its decreased regional level and other metabolic markers in comparison to controls. PET meta-analysis suggested that psychotic exacerbations are accompanied by immunological changes in microglia different from those seen in non-acute states and that the symptoms of schizophrenia can be modified by compounds such as non-steroidal anti-inflammatory drugs.	[142,143]
Deep brain stimulation (DBS)	Is mostly used to control Parkinson’s symptoms. This neurosurgical procedure is based on putting an electrode into the patient’s brain, and then connecting it to a pacemaker-like device. Based on this success, there is growing interest in using DBS to treat schizophrenia. DBS could target mostly the striatal dysregulation and is also considered for treating negative and cognitive symptoms.	[144,145]
Electroconvulsive therapy (ECT)	In this procedure, physicians induce a controlled epileptic seizure, which leads to some positive functional and structural changes in the CNS. That could explain the effectiveness of ECT in drug-resistant cases of schizophrenia.	[146,147,148]
Electrophysiological methods	There are many techniques to measure the electrical potentials of the CNS (evoked potentials, quantitative electroencephalography, mapping). In studies of patients with schizophrenia, these methods allow for the fundamental identification of inability to filter out (gating) useful from nonsensical information. However, linking electrophysiological results with cognitive disorders has so far turned out to be inconclusive.	[123,149,150]
Neurochemistry	There are lots of potential biochemical biomarkers of neuroplasticity in cerebrospinal fluid, blood, urine, and saliva. The key problem of clinical trials is the methodology of their measurements. It seems that the neurological effects of neuropeptides, like BDNF, can be monitored by the peripheral serum level.	[151,152,153,154]

## Data Availability

Not applicable.

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
