# Peer review of "Self-Regulatory Neuronal Mechanisms and Long-Term Challenges in Schizophrenia Treatment"

_brainsci, 2023, doi:10.3390/brainsci13040651_

Round 1

Reviewer 1 Report

In this paper, authors to evaluate the therapeutic effect of various rehabilitation methods, we reviewed methods that modify synaptic plasticity and improve cognitive and executive processes of patients with a diagnosis of schizophrenia. In this reviewe, PubMed, Scopus, and Google Scholar bibliographic databases were searched with keywords mentioned below. 555 records were identified. Modern methods of schizophrenia therapy with neuroplastic potential, including neurofeedback, transcranial magnetic stimulation, transcranial direct current stimulation, vagus nerve stimulation, virtual reality therapy, and cognitive remediation therapy, were reviewed and analyzed. If the authors complete minor revisions, the quality of the study will be further improved.

1. The introductory section is well written. If the authors describe in more detail the theoretical background related to self-regulatory neuronal mechanisms and long-term challenges in schizophrenia treatment in the introductory section, it will help readers understand.

2. line 177-185: Authors should describe more recent studies related to rTMS.

3. Authors should provide a richer description of the conclusion section.

Author Response

Dear Reviewer,

Thank you very much for reviewing our manuscript. We appreciate the interest and commitment you have provided for this work. We are very grateful for your extremely precious comments. We are convinced that thanks to your suggestions this manuscript will be much more valuable.

We wish you all the best!

Sincerely,
Agnieszka Markiewicz – Gospodarek
on behalf of all authors

Reviewer 2 Report

The text provides a clear summary of schizophrenia and its impact on cognitive and social functioning. It also highlights the limitations of current pharmacological treatments and the importance of exploring alternative treatments that target synaptic plasticity. The various methods reviewed, such as neurofeedback and cognitive remediation therapy, are described in enough detail to give the reader a basic understanding of each. The mention of individualized monitoring using methods like fMRI adds another layer of insight into the potential effectiveness of these alternative treatments. Overall, the text is concise, informative, and well-organized.

The text presents the limitation of pharmacological treatments for long-term treatment of schizophrenia and suggests that alternative treatments based on neural plasticity and self-regulation mechanisms should be considered. It also emphasizes the importance of monitoring the effectiveness of these treatments using methods such as fMRI and neuropeptides in serum for individualized treatment. Overall, the conclusions are supported by the information presented in the text.

The text does not specifically mention limitations, but some possible limitations that can be inferred from the information presented include:

  1. The limited duration of randomized controlled studies of long-term schizophrenia treatment, which do not exceed 2-3 years, may not fully capture the long-term effects of treatment.
  2. The availability and cost of some of the alternative treatments, such as neurofeedback and transcranial magnetic stimulation, may limit their accessibility for some patients.
  3. The effectiveness of these alternative treatments may vary across individuals and may not be effective for everyone.
  4. The potential adverse effects of some of the alternative treatments, such as transcranial magnetic stimulation and vagus nerve stimulation, may limit their use in some patients.
  5. The monitoring methods mentioned, such as fMRI and neuropeptides in serum, may not be widely available or accessible in all treatment settings.

Finally, the figures are not very clear and the table needs proper formatting. Also, the spacing is funny.

I am not a native english speaker but to me it seems OK

Author Response

(The authors gave the same response as above.)
